A plant-derived biostimulant Aminolom Enzimatico® application stimulates chlorophyll content, electrolyte leakage, stomata density and root yield of radishes under salinity stress

Kaya Gamze gamze.kaya@bilecik.edu.tr
Department of Horticulture, Faculty of Agriculture and Natural Sciences, Bilecik Şeyh Edebali University , Bilecik , Türkiye
Abd El-Moneim Diaa
Electronic publication date: 2025 Jan 24
Publication date: 2025
Volume: 13
Electronic Location ID: e18804
Received 2024 Sep 23; Accepted 2024 Dec 11
Copyright: © 2025 Kaya
Copyright year: 2025
Copyright holder: Kaya
License: This is an open access article distributed under the terms of the Creative Commons Attribution License, which permits unrestricted use, distribution, reproduction and adaptation in any medium and for any purpose provided that it is properly attributed. For attribution, the original author(s), title, publication source (PeerJ) and either DOI or URL of the article must be cited.
License URL: https://creativecommons.org/licenses/by/4.0/

Keywords: Raphanus sativus L., NaCl, Biostimulant, Electrolyte leakage, Stomata density

Funding: The authors received no funding for this work.

==============================
Biostimulants stimulate plant growth and tolerance to salinity stress, which creates unfavorable conditions for plant growth from emergence to harvest; however, little is known about their roles in triggering salt tolerance. Therefore, the study aimed to determine how applying a foliar plant-derived biostimulant (Aminolom Enzimatico® 24%) affects the growth (leaf area, biomass weight, root diameter, root fresh weight, and water-soluble dry matter), physiology (chlorophyll content, electrolyte leakage, cell membrane stability, and relative water content), and stomata of the lower and upper parts of leaves in radish plants (Raphanus sativus L.) under salinity stress. Radish plantlets at 7 d old were irrigated with saline water (i.e., 50, 100, 150, and 200 mM NaCl), and the biostimulant was sprayed twice at 7 d intervals. Under salinity stress, increased water-soluble dry matter content was detected, along with reduced plant biomass weight, root fresh weight, and root diameter. Meanwhile, the foliar biostimulant increased the mean root fresh weight, biomass fresh weight, and leaf area by 12%, 13.6%, and 24% compared to the control, respectively. Increasing NaCl dramatically reduced leaf area and relative water content, whereas chlorophyll content and stomata densities on both sides of the leaves improved. By regulating physiological parameters and thereby promoting root and leaf growth, the biostimulant application improved the radish plants’ tolerance to salinity stress up to 100 mM NaCl. Spraying the biostimulant can also boost plant growth, root yield, and radish quality under moderate salinity stress.

Introduction

High in vitamins and with mineral contents in their roots, radishes (Raphanus sativus L.) are among the most consumed vegetables in the world (Banihani, 2017). Radish cultivars come in several root shapes, colors, and sizes and, in Türkiye, are commonly preferred in dairy diets during the autumn and winter (Gunay, 2005). Given the rising popularity of consuming small radish cultivars, the plants are intensively cultivated in the Mediterranean region, especially in autumn and usually in open-field and greenhouse conditions where the salinity in soils and irrigation water sorely limits plant production (Colla et al., 2010; Jia et al., 2020; Khan, Baset Mia & Quddus, 2022). According to Maas & Hoffman (1977) and Sonneveld’s (1988) classification, radishes are a salt-sensitive or moderately salt-sensitive crop. Because radishes are frequently exposed to saline irrigation water, the adverse effects of salinity on the growth and yield of radishes have been investigated (Paromita et al., 2014; Garcia-Ibañez, Moreno & Carvajal, 2021; Ribeiro et al., 2024).

Salinity damage results from the toxicity of excessive salt ions and the water shortages created by the osmotic potential of salt ions (Munns & Tester, 2008). All stages of plant development from germination to flowering are negatively affected by salinity which disrupts ion balances, nutrient uptake, photosynthetic activity, and oxidative stress in plants, resulting in yield depletion (Jamil et al., 2007; Ayyub et al., 2016). A further consequence is the closure of the stomata, which limits the uptake of CO2 into the leaf tissue and consequently reduces the fixation and assimilation of carbon (Fricke, 2004). To lessen the adverse effects of salinity on plants, a wide range of commercial biostimulants have been developed and are now on the market, among which plant- and animal-based biostimulants have been recommended (Lisiecka et al., 2011). Because biostimulants include amino acids, enzymes, and humic substances, they encourage plant growth under various conditions of abiotic stress (Kauffman, Kneivel & Watschke, 2007), including drought (Celiktopuz et al., 2021) and both high (Repke et al., 2022) and low temperatures (Niu et al., 2022). Moreover, several studies have shown that plant-based biostimulants can enhance root growth, nutrient uptake, and photosynthesis under salinity stress (Ertani et al., 2013; Colla et al., 2014; Zuzunaga-Rosas et al., 2022). However, their specific effects on radish crops remain understudied. In a plant-specific case, Lucini et al. (2015) observed that a plant-derived biostimulant indeed enhanced the growth of lettuce plants exposed to salinity. Wang et al. (2022) and Shahzad et al. (2023) reported that biostimulants stimulated the photosynthetic activity of salt-stressed plants by increasing amino acids, chlorophyll content, antioxidant enzymes (SOD and CAT), and K+, Ca2+, Mg2+ concentrations and relative water content by regulating water uptake and loss. Biostimulants with high content of amino acids and active polypeptides enhanced root and shoot growth of maize under salt stress by regulating nitrate transport and reactive oxygen species (ROS) metabolism (Trevisan, Manoli & Quaggiotti, 2019) and led to increased nitrogen metabolism, osmolyte accumulation, and induced photosynthetic efficiency in lettuce (Lucini et al., 2015). Higher stomatal density was reported in broccoli plants subjected to biostimulants under salinity (Kaya, 2023). On the other hand, the efficiency of biostimulants on plant growth and root yield of radish has not been extensively studied by evaluating electrolyte leakage, leaf relative water content, and especially stomatal frequency, which regulates photosynthetic activities.

The study aimed to evaluate whether a plant-derived biostimulant Aminolom Enzimatico® with 24% amino acid contents alleviates the negative effects of irrigation with saline water on the growth, yield, quality, physiology, and stomata of radish plants.

Materials and Methods

Seedling growth conditions

A laboratory experiment was conducted in a plant growth chamber in Türkiye. Seeds of the radish (Raphanus sativus L.) cultivar Cherry Belle, with a vegetation period of 30–35 d, were sown in plastic vials and after 7 days, the uniform seedlings were transplanted to plastic pots filled with a 3:1:1 mixture of peat, perlite, and vermiculite; in that process, each plant was transplanted to its own pot. The pot size was 11 cm high with diameters of 10 cm at the top and 9 cm at the bottom. Throughout the experiment, the plants were grown in a cycle of approximately 16 h days (22 °C) and 8 h nights (18 °C) in 70% relative humidity.

Plant growth and salinity application

The pots were daily watered with distilled water until the beginning of irrigation with saline water, which began 7 d after transplantation. Irrigation salinities of 50, 100, 150, and 200 mM were constituted by NaCl solutions with electrical conductivities of 5.6, 11.2, 15.5, and 20.3 dS m−1, respectively. Meanwhile, irrigation with distilled water served as a control. All treatments were replicated four times, a total of 40 pots (4 salinity × 2 biostimulant × 4 replicates), and analyzed in a completely randomized design (CDR) Before the experiment, the plant growth medium was completely saturated with water, resulting in a relative water content of 100%. The treatment with saline water started when the relative water content dropped to 70% and it was maintained at around 85%. Every other day, each pot was weighed to determine the level of evaporation, and the amount of water lost was added with a respective amount of salt solution. A liquid form of Urea-P2O5-K2O (8-8-8 w/w) fertilizer diluted with 200 ml L−1 was applied as a starter after transplantation and 7 d later.

Biostimulant application

Eight pots were subjected to each salinity level, and four were sprayed with biostimulant twice at 7 d intervals. The plant-derived biostimulant Aminolom Enzimatico® 24% (35% organic matter, 16% organic carbon, 2% organic nitrogen, and 24% free amino acids; pH 4.5–6.5) was selected from the previous study conducted by Kaya (2023) in broccoli sprayed with a dose of 2.5 ml per liter water, as suggested for vegetables by the manufacturer, with Tween-20 (0.1%) added as an adhesive agent. A hand-held manual sprayer was used for foliar application, and 5 mL of biostimulant per plant was applied. Plants were harvested 15 days after biostimulant application.

Determination of morphological parameters

The plants were harvested 35 d after sowing and immediately washed with tap water. After the roots were wiped with a paper towel, and root traits (biomass fresh weight (root+leaves), root fresh weight, and root diameter) were recorded. All leaves from each plant were scanned to calculate leaf area per plant using Image J (Cosmulescu, Scrieciu & Manda, 2020).

Water-soluble dry matter

Water-soluble dry matter was determined in juice from the root of radish plants exposed to various salt levels and biostimulant application as a Brix unit (Atago Pal-1 Refractometer).

Determination of physiological parameters

Chlorophyll content measurement

The relative chlorophyll content was measured with a portable chlorophyll meter SPAD-502 (Konica Minolta Corporation, Osaka, Japan) as the SPAD index. After three successive measurements were taken in different locations of the leaves, mean values were calculated for each replicate.

Relative water content

To determine relative water content (RWC), fully expanded leaf from the top of plant was pulled off and directly weighed (FW). After they were immersed in distilled water in a falcon tube for 24 h in the dark at 20 °C, excessive water on the leaf surface was removed by paper towel and they were weighed to determine turgid weight (TW). At the end of imbibition period, the leaves were dried at a constant temperature of 80 °C for 24 h in an air oven (DW) (Kaya & Higgs, 2003). The following equation was used to determine RWC of leaves:

RWC(%)=[(FW−DW)/(TW−DW)]×100

Electrolyte leakage

Electrolyte leakage (EL) was determined by cutting the third leaf from the top of each plant and cleaning it with distilled water. After excessive water on the surface of the leaves was removed with paper towels, four discs 5 mm in diameter were excised, weighed, and subsequently soaked in 20 mL of distilled water in glass tubes, which were left for 24 h in an incubator at 20 °C in the dark. Electrical conductivity at time point 1 (i.e., EC1) was read using a WTW 3.15i conductivity meter at 25 °C. Afterward, the tubes were soaked in a thermostatic water bath at 90 °C for 40 min to kill all cells, at which point the electrical conductivity was again determined (i.e., EC2) at 25 °C after they were cooled in an incubator.

EL and relative injury (RI) were calculated using EC values as follows:

EL=(EC1/EC2)×100

RI(%)=[(ELs−ELc)/(100−ELc)]×100

in which, ELs and ELc are the EL for the salinity level and control sample, respectively (Gulen & Eris, 2003).

Stomatal density

The number of stomata and size were determined by an impression technique and the lower and upper surfaces of the third leaf from the top of each plant were used for detecting stomatal parameters. That leaf was carefully coated with transparent nail varnish in the area between the central vein and leaf edge and allowed to dry for 2–3 min. After drying, the varnish was gently removed from the lamina. The number of stomata per unit area was counted visually with a 40× objective lens and 10× eyepieces under a light microscope. Five samples from each field were randomly selected from different sections of each sample and counting was performed three replications. The size of the stomata in the photograph was also measured with an ocular micrometer connected to a stage micrometer and calculated using the stoma width and length in the following formula:

Stomasize(μm−2)=[(Stomatawidth/2)×(Stomalength/2)]×π

Photomicrographs were taken using a Zeiss Axiophot microscope with both image capture and digitalization program AxioVision 4.3.

Statistical analysis

Data were analyzed in a two-factor factorial in a completely randomized design with four replicates, with two plants in each replicate. Analysis of variance and a comparison of the means were performed by the JMP 13.0 statistical program and Tukey’s HSD test (p < 0.05), respectively.

Results

Analysis of variance revealed a significant difference in biostimulant and salinity levels, and their interaction was determined for all variables (Table 1). Root diameter, fresh weight, biomass fresh weight, and water-soluble dry matter were significantly higher in the plants treated with the biostimulant than in the control (Fig. 1). Except for water-soluble dry matter, those other traits were weakened by increasing the salinity.

Table 1 Analysis of variance and main effects of the biostimulant and salinity levels on plant growth parameters of radish.

Factor	Root diameter (cm)	Root fresh weight (g plant−1)	Biomass weight (g plant−1)	Water-soluble dry matter (Brix)	
Biostimulant (A)	
Control	27.5b	11.7b	16.8b	6.10b†	
Biostimulant	28.9a	13.1a	19.1a	6.70a	
Salinity stress (B)	
Distilled water	36.4a	23.1a	34.0a	3.51e	
50 mM	30.7b	14.6b	22.1b	5.45d	
100 mM	28.2c	10.9c	14.0c	6.25c	
150 mM	25.1d	7.9d	11.0d	7.60b	
200 mM	20.5e	5.7e	8.9e	9.08a	
Analysis of variance	
A	**	**	**	**	
B	**	**	**	**	
A × B	**	**	**	**	
Notes:

** Significant at 1%.

† Different letters indicate the significance level at p < 0.05.

Figure 1 The pictures of radish plants treated with the foliar biostimulant (upper row) under 0 (A), 50 (C), 100 (E), 150 (G), and 200 (I) mM NaCl and untreated (lower row) under 0 (B), 50 (D), 100 (F), 150 (H), and 200 (J) mM NaCl.

Morphological characteristics

Data regarding the two-way interaction are shown in Fig. 2. Root diameters were wider in radish plants sprayed with the biostimulant at all NaCl levels, although the positive effects disappeared under salinity exceeding 100 mM NaCl (Fig. 2A). A similar trend was observed in root fresh weight, which dramatically declined as the NaCl level rose (Fig. 2B). The beneficial effect of the biostimulant on root fresh weight was observed at salinity levels between 0 and 100 mM NaCl. Although each successive increase in salinity level led to a significant drop in biomass weight, the biostimulant spray promoted it considerably (Fig. 2C). Biomass weight in the radish plants sprayed with the biostimulant increased by 11.2%, 12.2%, and 14.7% at 50, 100, and 150 mM NaCl, respectively, compared with untreated plants. Water-soluble dry matter (Brix) also improved in the plants grown at different levels of NaCl, although no significant effect of the biostimulant was observed up to 150 mM NaCl (Fig. 2D).

Figure 2 Changes in root diameter (A), root fresh weight (B), biomass weight (C), and water-soluble dry matter (D) of radish plants treated with the biostimulant under salinity stress.

Bars and letters on each column show standard error (SE) and significance level at 5%, respectively.

Physiological characteristics

The primary and interaction effects of the biostimulant treatment and salinity levels were significant for all physiological parameters except chlorophyll content (Table 2). As expected, increasing salinity caused a decline in leaf area and RWC. Although applying biostimulant enhanced the leaf area, it decreased EL and RWC. A two-way interaction showed that chlorophyll content gradually increased with increased salinity levels and that the biostimulant’s effect was limited (Fig. 3A).

Table 2 Analysis of variance and main effects of biostimulant and salinity levels on physiological parameters of radish.

Factor	Chl content (SPAD)	Leaf area (cm−2)	Electrolyte leakage (%)	Relative water content (%)	
Biostimulant (A)		
Control	23.3	164b	45.1a	72.7a†	
Biostimulant	24.0	204a	41.7b	70.0b	
Salinity stress (B)		
Distilled water	22.1c	314a	13.2d	81.9a	
50 mM	23.5b	222b	46.5c	69.6b	
100 mM	23.5b	157c	57.3a	69.2b	
150 mM	23.6b	114d	54.7b	68.1b	
200 mM	24.3a	111d	45.3c	68.0b	
Analysis of variance		
A	ns	**	**	**	
B	**	**	**	**	
A × B	**	**	**	**	
Notes:

** Significant at 1%.

ns, not significant.

† Different letters indicate the significance level at p < 0.05.

Figure 3 Changes in chlorophyll content (A), leaf area (B), electrolyte leakage (C), and relative water content RWC (D) of radish plants treated with the biostimulant under salinity stress.

Bars and letters on each column show standard error (SE) and significance level at 5%, respectively.

Leaf area was significantly greater in the plants sprayed with the biostimulant at all NaCl levels but was reduced by salinity (Fig. 3B). The foliar biostimulant also lessened EL caused by salinity; however, the biostimulant’s positive effect disappeared at 150 and 200 mM NaCl (Fig. 3C). By contrast, in my study the RWC declined due to each increase in salinity (Fig. 3D). Under salinity stress, RI was lower in the plants sprayed with the biostimulant than the control. Although no significant difference was identified between the two groups of plants at 150 or 200 mM NaCl, a low RI was observed in the group treated with the biostimulant (Fig. 4).

Figure 4 Relative injury in leaves of radish plants treated with the biostimulant under salinity stress.

Bars and letters on each column show standard error (SE) and significance level at 5%, respectively.

The EL of treated and untreated radish plants was similar to each other in unstressed conditions but significantly increased under salinity stress. At the same time, it was lower in the biostimulant-sprayed plants than in the controls and significant decreases in EL were observed at 50 and 100 mM NaCl. Increasing salinity indeed reduced the RWC of leaves, and lower values were detected at 150 and 200 mM NaCl in the plants treated with the biostimulant (Fig. 3D).

RI peaked when salinity was 100 mM but was lower in plants sprayed with the biostimulant than the controls (Fig. 4). A promoter effect of biostimulant spraying was also observed up to 100 mM NaCl, although no significant improvement was determined at 150 or 200 mM.

Stomatal properties

A significant two-way interaction was identified in stomata density and size on the lower and upper epidermis of the radish plants (Table 3). The stomata density of the lower and upper epidermis improved as salinity levels increased, while the size of the stomata on the lower epidermis reduced under salinity stress (Fig. 5A). More frequent stomata were observed on the lower epidermis of the plants sprayed with the biostimulant at NaCl levels of 0, 150, and 200 mM (Fig. 6). Nevertheless, an increasing trend in stomata density on the upper epidermis was detected with increased NaCl, and the biostimulant application yielded higher stomata density than in the controls except at 100 mM NaCl (Fig. 5B). Larger stomata on both sides of the leaves was achieved in the plants treated with the biostimulant at all NaCl levels except 100 mM (Figs. 5C and 5D).

Table 3 Analysis of variance and main effects of biostimulant and salinity levels on stomata size and density of lower and upper epidermis of radish.

Factor	Stomata density (number mm−2)	Stoma size (µm2)	
Lower epidermis	Upper epidermis	Lower epidermis	Upper epidermis	
Biostimulant (A)	
Control	187b	116	255b†	243	
Biostimulant	199a	121	270a	249	
Salinity stress (B)	
Distilled water	165c	94d	305a	260	
50 mM	184b	113c	240c	228	
100 mM	185b	101d	256b	257	
150 mM	210a	137b	262bc	239	
200 mM	220a	148a	240c	248	
Analysis of variance	
A	**	ns	**	ns	
B	**	**	*	ns	
A × B	**	**	**	**	
Notes:

* Significant at 5%.

** Significant at 1%.

ns, nonsignificant.

† Different letters indicate the significance level at p < 0.05.

Figure 5 Changes in stomata density of lower (A) and upper (B) epidermis, and stomata size of lower (C) and upper (D) epidermis of radish plants treated with the biostimulant under different salinity levels.

Bars and letters on each column show standard error (SE) and significance level at 5%, respectively.

Figure 6 The pictures of stomata of radish plants treated with the biostimulant under different salinity stresses.

Relationship between the characteristics of radish

There were several significant correlations between the investigated traits in radish (Table 4). As expected, root fresh weight was related to biomass weight (r = 0.98**) and leaf area (r = 0.97**). There was a significant correlation between water-soluble dry matter and root diameter (r = −0.91**) as well as root fresh weight (r = −0.90**). Additionally, there was a positive correlation between water-soluble dry matter and leaf area (r = −0.83**). Electrolyte leakage was negatively correlated with leaf area (r = −0.85**) and biomass weight (r = −0.83**).

Table 4 Correlation coefficients between the investigated characteristics.

Corr. (r)	RD	RFW	BW	Brix	Chl	LA	EL	RWC	SNLE	SNUE	SSLE	
RFW	0.96**	–										
BW	0.93**	0.98**	–									
Brix	−0.91**	−0.90**	−0.87**	–								
Chl	−0.71**	−0.63**	−0.57**	0.78**	–							
LA	0.92**	0.97**	0.98**	−0.83**	−0.56**	–						
EL	−0.66**	−0.82**	−0.85**	0.63**	0.16	−0.83**	–					
RWC	0.70**	0.79**	0.78**	−0.76**	−0.36*	0.72**	−0.81**	–				
SNLE	−0.47**	−0.51**	−0.49**	0.65**	0.56**	−0.49**	0.38*	−0.46**	–			
SNUE	−0.68**	−0.67**	−0.63**	0.80**	0.77**	−0.62**	0.38*	−0.57**	0.72**	–		
SSLE	0.50**	0.47**	0.40**	−0.32*	−0.06	0.42**	−0.40*	0.39*	0.06	−0.11	–	
SSUE	0.14	0.13	0.08	−0.02	0.09	0.11	−0.15	0.10	0.19	0.08	0.66	
Notes:

*, **Significant at 5% and 1%, respectively.

RD, root diameter; RFW, root fresh weight; BW, biomass weight; Brix, water-soluble dry matter; Chl, chlorophyll content; LA, leaf area; EL, electrolyte leakage; RWC, relative water content; SNLE, stomata number on lower epidermis; SNUE, stomata number on upper epidermis; SSLE, stomata size on lower epidermis; SSUE, stomata size on upper epidermis.

Discussion

Root and shoot growth are the most valuable indicator characteristics for salinity stress because roots contact with the soil and transfer absorbed water to the shoot. Therefore, the reduction in root and shoot growth provides an important indication of the plant’s response to salt stress. A major effect of salinity is the reduction in biomass production and the inhibition of growth by limiting water uptake and ionic stress due to toxic ions (Munns & Tester, 2008). As shown in Fig. 2, a similar trend was determined in root diameter, root fresh weight, biomass fresh weight, and leaf area in radishes, which aligns with what Yıldırım, Turan & Donmez (2008) and Hussein & Joo (2018) research on radish has shown. However, root yield, biomass weight, and leaf area increased when the biostimulant was sprayed on the radish plants under salinity stress. The beneficial effects of biostimulants can result from the auxin-like effect of biostimulants, as well as enhanced nitrogen uptake and metabolism, regulation of the potassium-to-sodium ratio, and accumulation of proline, which serves as an osmoprotectant, protecting plants from salinity stress (Cerdán et al., 2013). Jamil et al. (2007) also found that salinity adversely affected the shoot weights, root weights, and leaf area of radish plants. By contrast, this study revealed that the foliar application of a plant-derived biostimulant could alleviate salinity’s adverse effects on the morphology of radish plants. Similar results have been reported by Yıldırım, Turan & Donmez (2008) in radish, as well as in lettuce (Lucini et al., 2015), and tomatoes (Turan et al., 2021; Zuzunaga-Rosas et al., 2022), and biostimulants derived from different sources improved plant growth under saline conditions in all those studies. It is supposed that increased plant growth in radish may be due to improved photosynthetic metabolism, which is consistent with the findings of Lucini et al. (2015), who found a higher photochemical activity (Fv/Fm) in lettuce plants sprayed with a plant-derived protein hydrolysate under salinity stress. Biostimulants’ foliar spray stimulates salinity tolerance in plants by regulating the accumulation of osmotic and antioxidant compounds that mitigate the adverse effects of salinity (El-Nakhel et al., 2022). The positive effects of Aminolom Enzimatico® on salt stress up to 100 mm appeared in radish because plants may have accumulated higher Na and Cl ions under high salt concentrations, which could cause imbalance, physiological drought, and destruction of biochemical processes. The plants subjected to salinity stress had higher chlorophyll content than control plants. Under salinity stress, the chlorophyll content is generally reduced because of the destroyed ion balance. In this study, the slight increase could be partly explained by the decrease in leaf area with a significant negative correlation (r = −0.56**). Also, reduced chlorophyll content under salinity is related to the plants’ tolerance threshold to salinity. Biostimulant application enhanced chlorophyll content under salt stress conductions by inhibiting chlorophyll degradation and inducing photosynthetic potential via amino acids, betaines, and mineral elements (Jafarlou et al., 2023). Previous research by Toscano, Romano & Patane (2023) found that one of three biostimulants had a positive effect on chlorophyll content in radishes. It is argued that the increase in chlorophyll content in radish plants may result from an increase in leaf nitrogen content and improved photosynthetic activity as reported by Colla et al. (2014), because Aminolom Enzimatico® contains 2% organic nitrogen and 24% amino acids stimulating chlorophyll contents.

Plants exposed to abiotic stresses show an increase in EL, which is a valuable indicator of stress tolerance levels. Under salinity, plant survival requires stable cell membrane integrity (Sanchez-Rodríguez et al., 2010; Hidalgo-Santiago et al., 2021). In this study, the application of Aminolom Enzimatico® reduced electrolyte loss from leaf tissues, suggesting it had protective effects against membrane damage. That finding corroborates the results of Jafarlou et al. (2023), who found lower EL in the plants treated with a biostimulant under moderate salt stress. This study revealed increased EL resulting from cell membrane damage in plants subjected to irrigation with saline water compared with the controls, whereas spraying the biostimulant supported the maintenance of membrane stability and reduced salinity’s hazardous effects on the radish plants. Also, it was negatively correlated with LA, RFW, and RWC, which are important marketable morphological characteristics.

Stomatal closure in plants subjected to salinity limits water loss by inhibiting transpiration (Siddiqui et al., 2021; Ahmad et al., 2021). Figueiredo et al. (2021) have demonstrated that applying a biostimulant attenuated salinity damage by improving photosynthetic capacity and reducing the transpiration rate in gooseberry plants. In the present study, stomata width increased density and decreased size were identified in the plants as salinity increased, which supports the results of Ahmad et al. (2021) regarding maize and Kaya (2021) regarding rockets. Thus, stomata size and density are valid indicators of tolerance to salinity stress in plants. Because their behaviors regulate photosynthetic activity and water loss by transpiration (Hidalgo-Santiago et al., 2021). However, stomata parameters were higher in the plants sprayed with the biostimulant than in the untreated plants under all salinity levels, thereby indicating that biostimulant-sprayed plants were less affected by salinity than the controls. Similarly, Kaya (2023) reported higher stomata density in broccoli plants sprayed with a plant-derived biostimulant under salinity. Beyond that, stomata closure or shrinkage induced a decrement in gas exchange and photosynthetic activity. Di Stasio et al. (2018) and Rouphael et al. (2017) similarly found that applying a biostimulant derived from seaweed improved gas exchange mostly by reducing stomatal closure. Also, increases in Na+ and Cl− contents in leaves resulted in a relative reduction in K content, leading to stomata closure under salinity stress.

Correlation analysis determined the relationship between the morphological and physiological characteristics of radish. Electrolyte leakage was negatively linked with leaf area, root fresh weight, and biomass weight. This suggests that increased electrolyte leakage caused damage to the integrity of the cell membrane, which could have adverse effects on the root yield of radish.

Conclusion

In sum, the application of Aminolom Enzimatico® 24% could mitigate the adverse effect of salinity stress on morphologic parameters and root yield by regulating physiological and stomata-related characteristics of radishes. The percentage reductions of root yield and biomass weight were lower in the plants treated with the biostimulant than in control plants. Although the radish plants showed sensitivity to salinity, the foliar biostimulant can improve tolerance to the adverse effects of salinity up to 100 mM NaCl by regulating EL, stomata density, and stomata size, thereby resulting in increased leaf area, root diameter, root, and biomass fresh weights. No significant improvement will be observed in biostimulant spraying under higher levels of salinity than 100 mM. Therefore, this study recommended the foliar application of biostimulant in radishes for improving the growth and yield under salinity stress conditions. Their application is simple, cheap, and does not need expensive and sophisticated equipment; consequently, they could be advised to farmers to alleviate salinity stress in radish production.

Supplemental Information

Supplemental Information 1 Raw data for statistical analysis.

Supplemental Information 2 Raw data.

Additional Information and Declarations

Competing Interests

The authors declare that they have no competing interests.

Author Contributions

Gamze Kaya conceived and designed the experiments, performed the experiments, analyzed the data, prepared figures and/or tables, authored or reviewed drafts of the article, and approved the final draft.

Data Availability

The following information was supplied regarding data availability:

The raw data are available in the Supplemental File.

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
