# Peer review of "A plant-derived biostimulant Aminolom Enzimatico® application stimulates chlorophyll content, electrolyte leakage, stomata density and root yield of radishes under salinity stress"

_PeerJ, doi:10.7717/peerj.18804_

## Round 0.1 · original submission · Major Revisions

Dear Authors

The manuscript cannot be accepted for publication in its current form. It needs substantial revision to meet the journal's standards. The authors are invited to revise the paper, considering all the suggestions made by both reviewers, including the reviewer who rejected the manuscript (R3). Please note that requested changes are required for publication.

With Thanks

Reviewer 1 ·

Basic reporting

The manuscript titled "Biostimulant Application Stimulates Physiological Parameters, Stomata Frequency, and Root Yield of Radishes Under Salinity Stress" offers significant insights into the role of biostimulants in alleviating salinity stress in radishes. However, several aspects of the manuscript require revision, as outlined below:
Keywords are missing. Please provide 5-7 keywords that must be unique than the title.
The text emphasizes salinity’s general effects on plant physiology and growth, but the discussion on radishes’ sensitivity to salinity could be more specific. The citation of Marcelis & Van Hooijdonk (1999) and other sources mention radishes’ response to salinity, but no detailed information on critical salinity levels for radish growth is provided. Including precise salinity thresholds (e.g., EC levels) and their direct effects on radish would strengthen the introduction. While the paragraph introduces biostimulants as a solution to salinity stress, the justification for their specific application in radish cultivation is weak. There is a significant gap between the mention of general biostimulant benefits (via studies on lettuce, maize, and broccoli) and their relevance to radishes. The authors should either reference studies directly involving radishes or explain why biostimulants used in other crops could be equally effective for radish under salinity stress. Instead of stating each study individually, group related findings and discuss their collective significance in the context of your study. For instance, "Several studies have shown that plant-based biostimulants can enhance root growth, nutrient uptake, and photosynthesis under salinity stress (Ertani et al., 2013; Colla et al., 2014; Zuzunaga-Rosas et al., 2022). However, their specific effects on radish crops remain understudied." Discuss the influence of salinity and bio stimulants on specific physiological traits or mechanisms being investigated, such as stomatal density, photosynthetic efficiency, or nutrient uptake, to better connect the reader to the study's objectives. Research gap is missing. Please clearly define the research gap.
Materials and Methods: The section mentions eight pots per salinity level with four receiving biostimulant, but more details on the replication process and randomization are needed. How Mention total number of plots used in experiment. Mention if all pots received equal light and temperature exposure, as these factors can introduce variability. Specify the total quantity of fertilizer applied per pot, and clarify whether all treatments received identical fertilization. How biostimulant spray was applied uniformly? Plants were harvested how many days after biostimulant spray?
In the results please ensure that all reported changes are explicitly linked to statistical significance. Statements such as "improved" or "increased" should be supported by p-values or a reference to Table 1 or 2 for clarity. Several key parameters in the results section lack quantification of percentage increases or decreases. To accurately assess the efficiency of the biostimulant, it is essential to incorporate these percentage changes for a more robust interpretation of its impact. The section reports several significant correlations between traits (e.g., root fresh weight and biomass, water-soluble dry matter and leaf area). However, the biological or agronomic relevance of these correlations is not explored. Include a discussion on the importance of these correlations. Clarify terms such as "water-soluble dry matter" and ensure consistency in the use of terminology throughout the results. Providing a brief definition or explanation in the methods or results section would prevent confusion.
In discussion, provide a more detailed, critical analysis of the results, emphasizing what distinguishes this study from prior work. Expand the discussion on the role of stomatal responses in plant adaptation to salinity stress. The two-way interaction effects between salinity and biostimulant treatments are mentioned, but their significance is not fully explored. For instance, the biostimulant showed beneficial effects up to 100 mM NaCl, but the reasons for the decline in efficacy at higher salinity levels (150 and 200 mM) are not discussed. Discuss potential mechanisms for why the biostimulant loses effectiveness at higher salinity levels. Include a more detailed discussion of the physiological mechanisms underlying the observed effects. If possible, draw from biostimulant chemistry (e.g., amino acids, betaines, minerals) to explain how these components might help plants manage ionic stress or maintain membrane integrity under saline conditions. This would help the reader understand the biological basis of the observed results. While the text states that the biostimulant's effect on chlorophyll content was limited, no detailed explanation or interpretation is provided for the observed increase in chlorophyll content under salinity. Provide a mechanistic explanation for the increase in chlorophyll content under higher salinity, considering that this finding contrasts with the general understanding of salinity stress. A literature reference supporting this phenomenon or a hypothesis explaining this anomaly would strengthen the interpretation. Discuss the physiological importance of reduced EL in greater detail. Explain how membrane stability (as indicated by lower EL) translates to improved stress tolerance and productivity. Similarly, elaborate on the significance of stomatal size and density in regulating gas exchange, water-use efficiency, and photosynthetic capacity under saline conditions. Refer back to the experimental data to strengthen these points.
Refine the conclusion to better reflect the complexity of the data and avoid overgeneralizing the results. Include the the practical applications of biostimulants in agriculture. Discuss how this research might inform broader practices in saline-prone regions or in crops with similar stress responses. Additionally, consider addressing limitations of biostimulant use, such as cost or the need for repeated applications, to provide a balanced perspective. Suggest future research directions.
Specific comments:
Line No. 24-26 “Under salinity stress, increased water-soluble dry matter content was detected, along with reduced plant biomass weight, root fresh weight, and root diameter.”
Line No. 26-28, “Meanwhile, the foliar biostimulant increased the 27 mean root fresh weight, biomass fresh weight, and leaf area by 12%, 13.6%, and 24%, 28 respectively.” while percentage changes are noted, these figures should be clearly tied to the specific conditions under which they were measured, i.e., was this improvement noted across all NaCl concentrations or only under specific conditions?
Line No. 28-29 “Increasing NaCl dramatically reduced leaf area and relative water content, whereas chlorophyll content and stomata densities on both sides of the leaves improved.” How and why chlorophyll content and stomatal densities increased under NaCl stress?
Line No. 38-44, The statement "Radishes (Raphanus sativus L.) are among the most consumed vegetables in the world" is generic and lacks specificity. While radishes may be important in certain regions, they are not universally a top-consumed vegetable. Instead, focusing on their regional significance or health benefits would provide a stronger context. Furthermore, the following line mentioning root shapes, colors, and sizes seems disconnected from the salinity stress topic and could be streamlined for better relevance.
There are redundant phrases and wordiness throughout the introduction, which can be streamlined for clarity and brevity. For example, (Line No. 40-41) "Given the rising popularity of consuming small radish cultivars, the plants are intensively cultivated..." Remove redundant information and unnecessary phrases to make the introduction more concise and readable.
Line No. 71-73, “Because radishes are frequently exposed to saline irrigation water, the adverse effects of salinity in enhancing the growth and yield of 73 radishes have been investigated” This phrase seems contradictory—salinity typically diminishes growth and yield, not enhances them.
Line No. 74-76, The statement "the study aimed to evaluate whether a plant-derived biostimulant foliar spray alleviates the negative effects of irrigation with saline water on the growth, yield, quality, physiology, and stomata of radish plants" is too broad. Present the study’s objective more concisely.
Line No. 35 “The plants were harvested d after sowing and immediately washed with tap water.” Plants were harvested how many days after biostimulant spray?
Line No. 106, “Eight pots were subjected to each salinity level, and four of them were sprayed with biostimulant 106 twice at 7 d intervals.” Given that eight pots were assigned to each salinity treatment, with four receiving the biostimulant spray and four without, the absence of data representing the salinity treatment with biostimulant (AXB) in the tables is concerning. The omission of these critical results raises questions about the comprehensiveness of the analysis and limits the ability to assess the standalone impact of salinity stress on plant growth and physiological traits. Clarification and inclusion of this data are essential for a complete understanding of treatment effects."
Line No. 173 “Photomicrographs were taken using a Zeiss Axiophot microscope” Mention the made and model of equipment.
Line No. 176, The statement that "data were analyzed in a two-factor factorial in a completely randomized design" is vague without further clarification of how the pots were randomized or blocked. Clarify whether pots were randomized within the growth chamber to avoid position effects, and explain how the biostimulant and salinity treatments were arranged in relation to each other.
Line No. 197-198, “Water-soluble dry matter also improved in the plants grown at different levels of NaCl”, What does the water soluble dry matter is actually indicating here?
Line No. 202-203 “As expected, increasing salinity caused a decline in leaf area, and RWC, but improved chlorophyll content.” Explain this phenomenon, how increased salinity can cause an enhancement in chlorophyll content?

Line No. 246-248 “Therefore, the reduction in root and shoot growth reduction provides an important indication of the plant's response to salt stress. Make correction in sentence by removing repeating words.

Experimental design

The experimental design lacks sufficient clarity. The authors are advised to address the ambiguities, particularly regarding the design structure and the timeline for plant analysis following the foliar application of the biostimulant. Providing these details will enhance the transparency and reproducibility of the study.

Validity of the findings

The tables currently do not present the results of the interaction between salinity and biostimulant treatments. Inclusion of these interaction effects is crucial for assessing the validity of the findings. Once the authors provide this data, a more accurate evaluation of the study's outcomes can be conducted.

Reviewer 2 ·

Basic reporting

Thank you for considering me for reviewing the manuscript “Biostimulant application stimulates physiological parameters, stomata frequency, and root yield of radishes under salinity stress”. The manuscript presents a valuable study on using biostimulants to alleviate the adverse effects of salinity stress on radishes. The research is timely and relevant, given the increasing need for sustainable agricultural practices in the face of climate change and soil salinization. The study focused on physiological and morphological parameters, which provide a general view of the biostimulant impact on radishes, making it a valuable contribution to the existing literature. However justification for using only one biostimulant product should be provided.

Suggestion:
Conduct a thorough English language edit of the manuscript to correct grammatical errors and improve clarity. Also, consider shortening lengthy sentences to enhance readability. Summarize redundant sentences and focus on highlighting the critical aspects of the study.

The abstract needs to be improved with more details on the methodology, particularly the applied biostimulant Aminolom Enzimatico. Consider streamlining by focusing on the most critical findings and their implications. Focus on the importance of the findings for growing Radishes under salinity stress.

Include relevant keywords that are not presented in the title and abstract to improve searchability.

The introduction requires significant improvement to enhance clarity, focus, and relevance. It could benefit from a deeper exploration of the mechanisms by biostimulants employing their effects under stress conditions and a more detailed review of recent literature. The justification behind choosing Aminolom Enzimatico as a biostimulant should be elaborated. It would be helpful to explain why this particular biostimulant was selected over other options and how its composition relates to its efficacy in alleviating salinity stress. In general the introduction should be reorganized to improve flow and clarity. Each section should smoothly shift into the next, allowing the reader to follow the logical progression of ideas. Discuss how this study builds on previous work or fills existing knowledge gaps. Communicate the knowledge gap, why this research is important, and what gaps in understanding it fills. Emphasizing these points will highlight the study novelty and make it clear to the reader why the research is timely and necessary. The hypothesis and objectives of the study should be clearly defined and presented in a separate, focused section towards the end of the introduction.

The methods are well-detailed and include information on growing conditions, salinity levels, and biostimulant application. Using appropriate control and treatment groups strengthens the experiment validity, and the data collection methods for measuring physiological and morphological parameters are clearly outlined.
But, justification for applying only one biostimulant product should be provided.
The experimental design is presented under the subsection of statistical analysis, and it should also be presented above in the experimental layout.

The results should be presented better in subsections. The resolution of Figures is currently inadequate, leading to unclear and pixelated images. These figures must be clear and legible for electronic publications at 100% zoom. Employing hierarchical clustering and heat map or PC biplot would be advantageous in exploring the relationships between the evaluated traits and the assessed treatments.

The discussion section is brief and greatly benefits from deeper analysis and insights. It lacks a clear, concise take-home message, it should focus more on interpreting the results within a broader context. A more detailed exploration of the physiological and biochemical mechanisms through which biostimulants enhance plant tolerance to salinity stress is needed. Additionally, the discussion should explain why the biostimulant effectiveness decreases at higher salinity levels, addressing the limitations of its application under more extreme stress conditions. Incorporating recent literature to support the findings and offering a more comprehensive revision would strengthen the discussion considerably. This will clarify the study contributions and relevance to scientific research and practical agricultural use.

Following the journal style guide or citation requirements is vital for uniformity and accuracy. Please review and standardize the reference section according to style guidelines. Revise the journal abbreviations in the references for consistency. Certain journals are abbreviated, such as J. Plant Nutr. (line 325) and Front Plant Sci. (line 328) while others are not such as Scientia Horticulturae (line 330,339 and 377) and Plant and Soil. (line 336 and 375). Ensure all journal titles throughout the references are correctly abbreviated according to the journal guidelines.

Experimental design

The experimental design is appropriate

Validity of the findings

The findings presented in the manuscript are valid, supported by robust data and sound statistical analysis.

Reviewer 3 ·

Basic reporting

See in section 4

Experimental design

ok

Validity of the findings

OK

Additional comments

The overall work was too much descriptive. The relationship about the results of different parameters was not discussed sufficiently. Most of the results were recorded and presented but did not provide reasonable science behind these results. The overall novelty of the work was very limited. There were many similar reports with similar approaches reported, thought the plants and testing parameters having slight difference.

---

## Round 0.2 · Minor Revisions

Dear Authors

The manuscript still needs a minor revision before publication. The authors are invited to revise the paper considering all the suggestions made by the reviewers. Please note that the requested changes are required for publication.

With Thanks

Reviewer 1 ·

Basic reporting

I appreciate the efforts made by the authors in incorporating most of the suggested revisions. However, a few important points remain to be addressed:
Several key parameters presented in the Results section lack quantification in terms of percentage increases or decreases. This omission limits the impact and interpretability of the findings. To enhance the analysis, it is recommended to include percentage changes for additional studied attributes, not just for biomass weight. Highlighting the most prominent percentage changes across these parameters will provide a clearer understanding of the magnitude of the observed responses, thereby improving the depth and clarity of the results.
The increase in chlorophyll content under higher salinity, as depicted in Fig. 3a without the application of a biostimulant, is unexpected given that salinity stress typically reduces these parameters. It would be beneficial to provide a mechanistic explanation for this finding, as it contrasts with the general understanding of salinity stress impacts on plant physiology. Supporting this observation with relevant literature or proposing a hypothesis to explain this anomaly would significantly strengthen the interpretation of the results.

Experimental design

The experimental design is well-structured and appropriate for addressing the research objectives.

Validity of the findings

The findings appear valid and well-supported; however, their full validity is contingent upon the author providing satisfactory responses to the queries highlighted in the Basic Reporting section. Addressing these concerns will ensure that the results are presented with the necessary clarity and detail, further enhancing the credibility and robustness of the study's conclusions.

Reviewer 2 ·

Basic reporting

The author has not completely addressed all previously mentioned comments.
The revised manuscript still lacks the recommended keywords.
The citations used in the revised version are outdated (e.g., Marcelis & Van Hooijdonk, 1999; Çavuşoğlu, Kılıç & Kabar, 2008) and need to be updated. Ensure that all citations throughout the manuscript are from studies published within the last ten years.
The experimental design is currently presented under the "Statistical Analysis" subsection; however, it should also be introduced earlier in the "Experimental Layout" section for clarity.
The results section would benefit from clearer organization by dividing it into subsections to enhance readability and flow.

Experimental design

The experimental design is appropriate

Validity of the findings

The findings presented in the manuscript are valid

---

## Round 0.3 · Minor Revisions

Dear Author
The manuscript still needs a minor revision before publication. The authors are invited to revise the paper considering all the suggestions made by the reviewer. Please note that the requested changes are required for publication.
With Thanks

Reviewer 1 ·

Basic reporting

I commend the authors for their thorough response to most of the suggested revisions. However, a significant anomaly requires further clarification:

The observed increase in chlorophyll content under higher salinity conditions, as reported in Table 2 and Figure 3a, is unexpected, particularly in the absence of a biostimulant. Salinity stress is typically associated with reduced chlorophyll levels due to its adverse effects on photosynthetic machinery. To better align this finding with established plant physiological responses to salinity stress, a mechanistic explanation is warranted.

Additionally, comparing chlorophyll content relative to leaf area could shed light on this anomaly. For example, under salt stress, the leaf area decreases (111 cm²), yet the chlorophyll content (24.3 SPAD) exceeds that of the control (leaf area = 164 cm²; chlorophyll content = 23.3 SPAD). This observation suggests that the increased chlorophyll concentration may result from a reduction in leaf area, leading to a denser chlorophyll distribution within the smaller leaf surface. A thorough discussion on the chlorophyll content-to-leaf area ratio in the context of previous literature would provide valuable insight into these results.

Experimental design

Experimental design is promising.

Validity of the findings

The findings presented are valid; however, I recommend that the authors give more detailed consideration to the suggestions in the basic reporting section, particularly regarding the physiological parameters. A deeper focus on these aspects will enhance the clarity and interpretive strength of the study.

---

## Round 0.4 · Major Revisions

Dear Author

The manuscript cannot be accepted for publication in its current form. It needs a major revision before publication. The authors are invited to revise the paper considering all the suggestions made by the reviewer. In addition, there are more comments:

1) The title is too vague: Which biostimulant was used, and which physiological parameters were affected? Is it stomata frequency or density?
2) In addition, the writing must be improved. Starting with the very first sentence: "Biostimulants have great attention in recent years for stimulating plant growth and tolerance to salinity stress, which creates unfavorable conditions for plant growth from emergence to harvest; however, little is known about their roles in triggering salt tolerance": Phrases such as "have great attention in recent years " are just filling without contributing information. Authors should try to be more concise and specific.

Please note that the requested changes are required for publication.
With Thanks

Reviewer 1 ·

Basic reporting

Author incorporated most of the suggested revisions. Now the article can be accepted for publication.

Experimental design

Experimental is explained with sufficient details for reproduciability.

Validity of the findings

Results are valid and promising.

Reviewer 4 ·

Basic reporting

The research is original and novel. The research needs more clarity. The language was not correctly used by author, as presented by many grammatical mistakes. The introduction and review is aligned with the study, however, references are quite old that can be replaced with the recent studies. Figures are relevant to the study and high-resolution pics were prepared. The labelling of figures is appropriate; please clearly define the study rationale and research gap. Please discuss how different salinity levels and relative water content were maintained? Properly describe whether salinity is the real problem in radish. Properly discuss the results and trace a relationship between different parameters. Also, discuss the future perspective and any limitations of the study in the conclusion section. Please write Title in sentence case.

Experimental design

The experimental design is correctly used to meet the objectives. However, details are not given properly. From where the seeds were collected and how the salinity levels were maintained. How the ratio of biostimulants were maintained for different treatments. Experimental setup photo may please be used

Validity of the findings

The experiments have different flaws. The results are not written properly. There is a lack of comparison with latest studies. The percentage shall be linked to the increase/decrease in salinity (or to cover the adverse symptoms of salinity).

Additional comments

Salinity is a major problem across the globe, however the authors shall mention the real figures that how much percentage of radish is lost in Turkey due to salinity. Author shall mention the statistical figures of last year to fill the research gap. Grammar is weak and the results are not clearly mentioned. References need formatting.

---

## Round 0.5 · Minor Revisions

Dear Authors
The manuscript still needs a minor revision before publication. The author is invited to revise the paper considering all the suggestions made by the reviewers. Please note that the requested changes are required for publication.
With Thanks

Reviewer 2 ·

Basic reporting

The author has not addressed previously mentioned comments.
The revised manuscript still lacks the recommended keywords.
The citations used in the revised version are outdated (e.g., Marcelis & Van Hooijdonk, 1999; Çavuşoğlu, Kılıç & Kabar, 2008) and need to be updated. Ensure that all citations throughout the manuscript are from studies published within the last ten years.
The experimental design is currently presented under the "Statistical Analysis" subsection; however, it should also be introduced earlier in the "Experimental Layout" section for clarity.
The results section would benefit from clearer organization by dividing it into subsections to enhance readability and flow.

Experimental design

The experimental design is appropriate

Validity of the findings

The findings presented in the manuscript are valid

Reviewer 4 ·

Basic reporting

Suggested changes in previous review have been incorporated

Experimental design

The author have incorporated majority of the suggestions

Validity of the findings

The findings are valid
The author have incorporated majority of the suggestions

Additional comments

The paper may please be accepted as author have incorporated majority of the changes

---

## Round 0.6 · accepted · Accept

Dear Authors,

I am pleased to inform you that the manuscript has improved after the last revision and can be accepted for publication.

Congratulations on accepting your manuscript, and thank you for your interest in submitting your work to PeerJ.

With Thanks


Reviewer 2 ·

Basic reporting

The authors have thoroughly addressed all the comments provided, incorporating the necessary revisions and improvements to enhance the clarity of the manuscript.

Experimental design

The experimental design is appropriate

Validity of the findings

The findings presented in the manuscript are valid